# Effect of Photobiomodulation Combined with Physiotherapy on Functional Performance in Children with Myelomeningo-Cele-Randomized, Blind, Clinical Trial

**DOI:** 10.3390/jcm12082920

**Published:** 2023-04-17

**Authors:** Tamiris Silva, Daysi da Cruz Tobelem, Tainá Caroline Dos Santos Malavazzi, Juliana Fernandes Barreto de Mendonça, Lucas Andreo, Maria Cristina Chavantes, Anna Carolina Ratto Tempestini Horliana, Karina Helga Leal Turcio, Andréa Oliver Gomes, Alessandro Melo Deana, Kristianne Porta Santos Fernandes, Lara Jansiski Motta, Raquel Agnelli Mesquita-Ferrari, Aldo Brugnera, Samir Nammour, Sandra Kalil Bussadori

**Affiliations:** 1University Nove de Julho (UNINOVE), São Paulo 01525-000, SP, Brazil; 2University of Canberra, Bruce, ACT 2617, Australia; 3Sao Paulo State University (UNESP), Aracatuba 11330-900, SP, Brazil; 4Physics Institute of São Carlos, University of the São Paulo, São Paulo 11330-900, SP, Brazil; 5University of Liege, 4000 Liege, Belgium

**Keywords:** myelomeningocele, physiotherapy, photobiomodulation, functional ability, surface electromyography

## Abstract

Background: This study aimed to evaluate the electrical activity of the rectus femoris, tibialis anterior, and lateral gastrocnemius muscles during the sit-to-stand task and functional mobility after a neurofunctional physiotherapy protocol associated with PBM. Methods: Twenty-five children were randomly allocated to either Active PBM + physiotherapy (n = 13) or PBM sham + physiotherapy (n = 12). PBM was carried out with a LED device (850 nm, 25 J, 50 s per point and 200 mW) at four points over the area with absence of a spiny process. Both groups completed a twelve-week supervised program with two weekly 45–60 min sessions. Pre-training and post-training assessments involved the Pediatric Evaluation of Disability Inventory (PEDI). Muscle activity was assessed using portable electromyography (BTS Engineering) and the electrodes were positioned on the lateral gastrocnemius, anterior tibialis, and rectus femoris muscles. The RMS data were recorded and analyzed. Results: After 24 sessions of the treatment protocol, improvements were found in the PEDI score. The participants presented greater independence in performing the tasks, requiring less assistance from their caregivers. More significant electrical activity was found in the three muscles evaluated between the rest period and execution of the sit-to-stand tasks, both in the more compromised or less compromised lower limbs. Conclusion: Neurofunctional physiotherapy with or without PBM improved functional mobility and electrical muscle activity in children with myelomeningocele.

## 1. Introduction

Myelomeningocele (MMC) is the most severe form of spina bifida [1]. Although the disease etiology remains unclear, myelomeningocele is thought to be multifactorial, involving an interplay of environmental and maternal factors, with a global incidence of one case in every 1000 live births [2].

Myelomeningocele is a neural tube defect that occurs during embryonic development due to incomplete closure of the spinal neural tube. It ultimately leads to an exposed neural tissue or meninges with a fluid-filled sac that protrudes at the affected vertebral level. The unprotected neural tissues suffer progressive harm due to exposure to chemical and mechanical factors of the intrauterine environment [3]. Individuals with MMC often present motor, sensory, and neurological deficits below the lesion level, which can result in lower limb paralysis or weakness, skeletal deformities, sensory deficit, bladder and bowel incontinence, and hydrocephalus. These impairments may impact the child’s functionality, placing them at risk for decreased social participation [4,5].

Most children with MMC take longer to learn how to control their lower limbs enough to acquire basic motor skills, such as sitting, crawling, and walking [6]. Schoenmakers et al. [7] suggest that the level of the lesion, cognitive impairment, muscle weakness, and contractures in the lower limbs are essential determinants of functional independence regarding mobility. Although the neurological level of the lesion is a strong predictor of walking status in children with MMC, the authors described that some modifiable factors could contribute to functional independence, such as the strength of the knee extensors and range of motion of the lower limb joints [7].

Individuals with MMC have functional limitations and often need assistance to perform activities of daily living, which can lead to a negative impact on quality of life [8,9]. Emerging evidence suggests that physical activity may positively impact the development of the central nervous system in children with both typical and atypical development [9]. Early interventions for children with MMC should involve structured strategies to facilitate the development of motor skills to reduce the severity of the disability in adolescence and adulthood [9]. However, there is now a clear dichotomy in the evidence base for what does and does not work for improving function and performance of tasks in this population.

Therefore, increasing physical activity for children with MMC may be critical since a loss of strength and mobility may lead to less independence and functionality to carry out activities of daily living. Physiotherapy is an important feature of the rehabilitation of individuals with MMC. However, knowledge of which physiotherapeutic strategies should be used or the best frequency and intensity of the sessions are still lacking [9].

Therapeutic exercises should be functionally performed for the reorganization and adaptation of new circuits in the central nervous system to be more influential within the recovery process [10] When an individual is learning a new function, this connectivity process is very important. At the beginning of learning a new motor skill, more areas at the cortical level are activated as well as a greater energy expenditure. Therefore, repetitions play an essential role in movement learning [11]. As a child experiences trial and error to acquire the new motor skill, the new function begins to be properly learned, and these circuits are managed and optimized. An important factor at this stage is the movement’s influence on the activity to become automatic. Therefore, when selecting therapeutic interventions, we should consider not only the level of affected structures (joint, muscular, and sensory components), but also the context in which the child is embedded [12,13]. During the intervention planning process, it is essential to include functional activities that make sense to the child [14,15].

The current literature describes therapeutic approaches that utilize light sources to assist in rehabilitating patients with neurological diseases such as stroke, neurodegenerative diseases, and spinal cord injury. Photobiomodulation (PBM), in turn, involves the application of low-intensity light sources, such as low-level laser therapy or light-emitting diodes (LEDs) to biological tissues [14,15].

It was initially believed that the coherence of LBI light was essential to achieve therapeutic effects. However, therapeutic benefits were also found using LED, an incoherent light source [16]. LBI and LED emit monochromatic radiation, although LED has a wider spectral width. LBI waves have an organization that favors greater collimation. In contrast, the LED waves are incoherent and the light is therefore not collimated, allowing for treatment of larger areas. Despite differences in their mode of operation, both devices are efficient for FMB. Nevertheless, LED has the advantage of cost-effectiveness [16,17].

There is currently no evidence on the use of PBM for the treatment of MMC. However, studies utilizing experimental models of spinal cord injury have shown PBM to be a potentially effective and non-invasive therapy, promoting axonal sprouting. An increase in the glial cell A clinical trial involving individuals diagnosed with spinal cord injury revealed that PBM positively affects motor function, particularly during the isotonic contraction of stimulated muscles, as evaluated by electromyography (EMG) [18]. Moreover, Silva et al. [19] have indicated that after 12 PBM sessions associated with physiotherapy in spinal cord injury patients, there was a recovery in sensory perception and muscular strength. These promising results suggest that PBM could be an effective treatment for promotion of functional and sensory improvements.

Therefore, this study aimed to evaluate the electrical activity of the rectus femoris, tibialis anterior, and lateral gastrocnemius muscles during the sit-to-stand task and functional mobility after a neurofunctional physiotherapy protocol associated with PBM (compared with a group that underwent a neurofunctional physiotherapy protocol and sham PBM). The hypothesis of the present study is that the neurofunctional physiotherapy exercises associated with PBM would increase the muscles’ muscular electrical activity and consequently improve the functional mobility of children diagnosed with MMC.

## 2. Materials and Methods

This is a clinical, controlled, randomized, and blind trial. It complies with regulatory standards (resolution number 466/2012) for research in human beings and has approval from the Research Ethics Committee of University Nove de Julho (number: 4,308,134). It was registered with Clinical Trials (ClinicalTrials.gov Identifier: NCT04425330). Consolidated standards for clinical trial reporting (CONSORT) were followed to ensure study quality and transparency.

Eligible participants and their legal guardians were informed about all details of the study. Subsequently, legal guardians signed a statement of informed consent, while children signed a term of assent. The researchers used images to explain the procedures and then placed paint on the child’s finger to mark the “yes” response (“I want to participate”) or “no” response (“I do not want to participate”). The participants were submitted to evaluations, the results of which were recorded on the initial evaluation form.

### 2.1. Participants

Inclusion criteria: (i) age 2 to 12 years; (ii) diagnosis of MMC on the lower lumbar and sacral level; (iii) able to perform sit-to-stand movement with or without assistance.

The exclusion criteria were as follows: (iv) cognitive impairment affecting capacity to communicate and understand (por exemplo, não entender o commando para se levanter e sentar durante); (vi) manifestations secondary to MMC (e.g., tethered spinal cord syndrome, syringomyelia) (vi) neuromuscular scoliosis; (vii) hip and knee luxation; (viii) other central nervous system disease (e.g., cerebral palsy).

### 2.2. Randomization

Eligible children were randomly allocated to either the Physiotherapy + LED group (Active LED) or the Physiotherapy + sham LED group (Sham LED) for 12 weeks. Randomisation for the study was provided through a randomization website, accessed on September 2019 (randomization.com) to ensure concealed allocation.

### 2.3. Experimental Design

The participants were selected based on eligibility criteria and were randomly allocated to either the Active LED group (n = 10) or the Sham LED group (n = 10). Both guardians were blinded to group allocation. Before and after the training protocol, the participants underwent measures of body structures, functional mobility, and caregiver assistance. All children commenced training the week after baseline assessment. Intervention period lasted 12 weeks and involved two weekly 40–60 min sessions. Assessments were conducted prior to the treatment protocol (T0—pre-intervention), after 24 sessions (T1—post-intervention), and 30 days after completion of intervention (T2—follow-up).

### 2.4. Intervention

#### Neurofunctional Physiotherapy

Neurofunctional physiotherapy was performed using functional activities and active movements. After baseline assessment, individual intervention goals were defined by the participants or their family member (e.g., walking without assistance during recess at school) using the F-Words goal sheet [20].

Both groups performed 45–60 min of structured-play-based physical activities twice a week for 12 weeks and aimed to strengthen the lower limb and trunk muscles, improve balance, and stimulate sensory and perception systems. These activities involved:Hanging positions;Squatting to pick up objects from the floor;Going up and down steps;Sitting down and standing up from a chair;Changing positions from kneeling to semi-kneeling and standing;Walking with or without assistance around obstacles, and stable and unstable surfaces.

During these activities, the participants wore ankle weights (0.5 to 1.0 kg) to increase strength and body self-perception. The participants also engaged in ludic activities that enabled them to perform tasks associated with the intervention goals.

Trunk control and muscle strengthening were performed using functional movement patterns, such as changing from the lying to the sitting position and with balls and rolls. Isolation exercises were also performed to strengthen hamstrings, quadriceps, and gluteus medius muscles. These exercises were performed with repetitions (one set of 10 repetitions) to promote motor learning and enhance lower limb strength and endurance [21,22,23].

### 2.5. Photobiomodulation Protocol

The participants were comfortably positioned in the prone position. Palpation of the spiny processes was performed to locate the defect and determine the irradiation site. The site with the absence of a spiny process was considered the location of the defect. Four points of this site were then irradiated (Figure 1).

PBM was administered using a prototype comprised of light-emitting diodes (LED) at a wavelength of 850 nm connected to an energy source (MPS-3005B-MINIPA, São Paulo, SP, Brazil). The same device was used in both groups. However, the device does not emit light in sham mode (sham LED group) (Figure 2).

The power of the equipment was measured using the FieldMaxII-TO power meter (COHERENT^®^) to ensure no loss of energy. Twenty-four sessions were held at a frequency of twice per week, totaling 12 weeks. The irradiation parameters are listed in Table 1.

All interventions were delivered individually, and to ensure protocol adherence, therapists were trained by the research lead (TS) on intervention delivery and their outcomes. The therapists were responsible for recording interventions, in particular, type, intensity, repetitions, and performance. The intervention was discontinued if the participant missed three non-consecutive sessions.

### 2.6. Assessments

#### 2.6.1. Functional Mobility and Caregiver Assistance

The Pediatric Evaluation of Disability Inventory (PEDI) was used to assess functional independence. This questionnaire has been translated to Portuguese and culturally adapted to address Brazilian sociocultural specificities [8]. The questionnaire was administered in interview form to the parents or caregivers of the children and was comprised of three parts: I—functional abilities (197 items); II—caregiver assistance (20 items); and III—modifications to the environment (20 items). Part I addresses the child’s ability to perform daily activities and tasks. Part II addresses the quantity of assistance that the caregiver must provide for the child in order to perform daily activities and tasks. Part III addresses the quantity of modifications to the environment used by the child. Each part is subdivided into three function domains: self-care, mobility, and social function [9].

The mobility domain of part I was used in the present study, which is composed of 65 items. Each item was scored as either 0 (unable to perform task) or 1 (able to perform task). The total is obtained by the sum of the item scores. For the caregiver assistance portion, the items were scored on a scale of 0 (completely dependent) to 5 (completely independent). The caregiver assistance portion of the mobility domain of part II was also evaluated, investigating the quantity of assistance provided by the caregiver for the execution of the evaluated tasks in the functional mobility domain. Greater help required from the caregiver to perform functional activities denotes a lower degree of independence. A score of 5 was attributed if the child could perform the tasks independently and a score of 0 was attributed if the child required complete assistance. Intermediate scores range from 4 to 2. The PEDI was scored in accordance with the instruction manual. [23]

#### 2.6.2. Electrical Muscle Activity

Electrical muscle activity was determined using a portable electromyograph (BTS Engineering, Milan, Italy) with a bioelectrical signal amplifier, wireless transmission, and bipolar electrodes with a total gain of 2000× and sampling frequency in the range of 20 to 450 Hz synchronized to the BTS EMG-analyzer. The sit-to-stand (STS) task was used as the measure of functioning [24,25]. The electrodes were placed on the motor point of the lateral gastrocnemius, tibialis anterior, and rectus femoris muscles [24,25]. The muscles were measured with a metric tape to ensure the placement of the electrode on the motor point according to the SENIAN guidelines for noninvasive electromyography (EMG) assessments [26]. EMG data were filtered using the EMG-Analyzer (BTS Engineering, Milan, Italy). Root mean square (RMS) values were extracted and analyzed.

A standard method for representing the intensity or amplitude of an electromyographic signal is through the root mean square (RMS), which is an average calculated by a computerized electromyography program that represents voltage over a study cycle. The RMS is calculated as the root mean square of the input values. Specifically, if N represents the number of samples and xi represents the value of each sample, the RMS value can be defined as:$$RMS=\sqrt{\frac{1}{N}}\begin{array}{c} N-1\\ \sum \\ i=0 \end{array}x^{2}i$$

For the STS task, the participant was positioned on a chair with hip, knee, and ankle flexed at 90° and feet supported on the floor without the use of braces. When necessary, an evaluator positioned the feet.

Electrical muscle activity was first determined with the child sitting at rest. The child was then asked to stand, remain in the standing position for ten seconds, and then sit down again. The sit-to-stand task was performed three times with a five minute interval between trials. The children performed the movement at a pace usually adopted in their daily routine. Collection time was approximately 30 min, always with rest intervals between readings to avoid the effects of fatigue.

### 2.7. Sample Calculation

The sample size was calculated based on the outcome of the PEDI scale in the study conducted by Aizawa et al. [25], considering a 95% confidence level. The result was 30 individuals divided into two groups with an effect size of 0.8 and test power of 0.9566, maintaining the significance level at α = 0.05.

### 2.8. Statistical Analysis

Data were tabulated and treated using the GraphPad PRISM program, version 8.0. The Shapiro–Wilk test was used to determine normal distribution of continuous data. Data with Gaussian distribution were expressed as mean and standard deviation. Nonparametric data were expressed as median and range (minimum, maximum). Comparisons between groups were performed using two-way repeated-measures ANOVA for parametric data. Pairwise comparisons were performed using the paired *t*-test for parametric data and the Wilcoxon test for nonparametric data, corrected by the Ryan–Holm–Bonferroni decreasing procedure. The level of significance was set at α = 0.05.

## 3. Results

Forty-two volunteers were screened for eligibility between October 2020 and September 2021. Sixteen children were excluded. Twenty-five participants were randomized and allocated into two groups. However, five children (three from the Active LED group and two from the Sham LED group) withdrew from the study. Thus, 20 children completed the study (10 in each group) (Figure 3). Characteristics of participants are presented in Table 2. No differences between groups were found regarding age or lesion level (*p* > 0.05, paired *t*-test).

### 3.1. Functional Mobility and Caregiver Assistance

In the analysis of functional mobility, no difference was found between groups prior to the intervention (T0) (*p* > 0.05, Wilcoxon test). After 24 sessions of the treatment protocol (T1), improvements were found in both groups (*p* < 0.01, Wilcoxon test). Moreover, the participants presented greater independence in performing the tasks, requiring less assistance from the caregiver (*p* < 0.01, Wilcoxon test) (Figure 4).

### 3.2. Electrical Muscle Activity

For the analysis of electrical muscle activity, the limbs were divided into less compromised and more compromised. The RMS data were extracted and analyzed. Significant differences were found in the three muscles evaluated between the rest period and execution of the sit-to-stand tasks at both T0 and T1 in both treatment groups and in both the more compromised and less compromised limbs (*p* < 0.05, two-way ANOVA). No significant differences between the Physiotherapy + LED and Physiotherapy + sham LED groups were found at T0 (*p* > 0.05, unpaired *t*-test).

### 3.3. Compromised Lower Limbs

#### More Compromised Lower Limbs

Data analysis showed greater electrical activity of the rectus femoris muscle during the STS task at T1 compared with T0 in both the physiotherapy + LED group (*p* = 0.004, paired *t*-test) and physiotherapy + sham LED group (*p* < 0.001, paired *t*-test). An improvement was found in the electrical activity of the tibialis anterior muscle at T1 compared with T0 in both the physiotherapy + LED group (*p* =< 0,05, paired *t*-test) and physiotherapy + sham LED group (*p* < 0.001, paired *t*-test). Likewise, a significant improvement was found in the electrical activity of the lateral gastrocnemius at T1 compared with T0 in both the physiotherapy + LED group (*p* < 0.0001) and physiotherapy + sham LED group (*p* = 0.0023, paired *t*-test) (Figure 5).

### 3.4. Less Compromised Lower Limbs

Data analysis showed that a significant improvement was found in the electrical activity of the rectus femoris muscle during the STS task at T1 compared with T0 in both the physiotherapy + LED group (*p* < 0.01, paired *t*-test) and physiotherapy + sham LED group (*p* < 0.001, paired *t*-test). A significant improvement was found in the electrical activity of the tibialis anterior at T1 compared with T0 in both the physiotherapy + LED group (*p* < 0.001, paired *t*-test) and physiotherapy + sham LED group (*p* < 0.001, paired *t*-test). Likewise, a significant improvement was found in the electrical activity of the lateral gastrocnemius at T1 compared with T0 in both the physiotherapy + LED group (*p* < 0.001, paired *t*-test) and physiotherapy + placebo LED group (*p* < 0.001, paired *t*-test) (Figure 6).

## 4. Discussion

The present study is the first to investigate the effects of neurofunctional physiotherapy combined with PBM in children with MMC. Our results revealed that the proposed exercise protocol improved functional mobility and diminished the need for assistance from a caregiver during the mobility task evaluated using the PEDI and increased muscle activity during the STS task, regardless of active or sham PBM.

According to Lee et al. [9], the first years of life are considered a critical period for motor development, offering the best opportunity for the introduction of interventions that facilitate the development of motor skills in children with MMC. Interventions initiated early and with high intensity offer better and longer lasting results in infants with atypical development compared with those who began later and at lower intensities [9].

Therefore, only one study has reported the effects of physiotherapy in children with MMC [13]. Aizawa et al. [23] investigated whether children with MMC improved motor ability and functional independence after treatment consisting of ten weekly 45-min sessions of conventional physiotherapy compared with a physiotherapy program based on reflex stimulation. The conventional physiotherapeutic interventions centered on muscle strengthening, the maintenance of a position for the longest time possible (sitting, crawling, kneeling, and standing), and changes of position (rolling, transition from supine position to sitting, from sitting to crawling, and from crawling to kneeling). The study showed that the group that received physiotherapy with reflex stimulation performed myotatic reflexes with muscle stretching before and during contraction of the muscle or by percussion of the tendon. The passive and assisted phase of the rhythmic initiation stimulated rolling, sitting, and crawling. Manual assistance was performed in at least two muscles or muscle regions with five repetitions minimum in each session. Both groups presented improvements in gross motor function, self-care, and mobility domains of the PEDI. These findings align with our results, in which the participants of both treatment groups presented improvements in functional mobility, accompanied by an increase in functional independence, requiring less assistance from a caregiver.

A more recent systematic review conducted by Novak et al. in 2019 [27] showed effective interventions for the motor rehabilitation of children with cerebral palsy. According to the authors, training-based interventions focused on activities of daily living that involve active and self-generated movements can improve functional mobility in children with cerebral palsy. Functional mobility training was shown to be feasible and effective across children with cerebral palsy but has received little attention in children with MMC. The potential benefits of such training could also improve functional mobility in children with MMC. Therefore, the current understanding of the evidence for effective motor interventions in MMC is of great importance.

The physiotherapy sessions in the present study focused on functional activities with changes in position combined with play and strengthening exercises focused on lower limbs and trunk muscles. The results of this study are promising. Besides the improvements observed in functional mobility and the reduced need for caregiver assistance, our study shows an increase in electrical muscle activity in areas considered vital for gait [28]. These results are similar to those reported by Khan et al. [29], who found that the specific training of tasks and repetitive exercises are important factors for synaptogenesis and essential elements for acquiring motor skills. Thus, both time and dose (intensity and frequency) of the treatment protocol may be important and generate opportunities for practice beyond the clinical setting.

The ability to rise unassisted from a seated position is one of the most fundamental movements among activities of daily living. The STS movement is an essential precursor to independence in everyday life, such as walking. There are necessary components to completion of the STS task, including the progression of desirable directions and generated force, the ability to maintain stability during the execution of the task, and the adaptation of the position or movement when the environment is altered [30]. Previous studies with adults revealed that the joint movements of the two lower extremities were asymmetrical during the execution of the STS task [31]. This asymmetry may be due to the different actions of the dominant and non-dominant lower limbs.

### Improvements in Electrical

Our study shows that physiotherapy with or without PBM may improve the electrical activity of the rectus femoris, tibialis anterior, and lateral gastrocnemius muscles in both groups and in the more compromised and less compromised limbs. Moreover, the treatment induced the independent walking ability of 11 of the 20 children. There is corroborative evidence that lower limber strength patterns may predict the ambulation of children with MMC. A study conducted by McDonald et al. [32] showed that strength of the iliopsoas, quadriceps, tibialis anterior, and gluteus muscles are considered ambulation predictors in children with MMC. Similarly, Schoenmakers et al. [7] reported that knee extensors seem to be significantly associated with mobility independence, including changes of position, walking in internal and external environments, and climbing stairs. The authors also stated that the level of the lesion may be less important than the strength of the lower limbs with regard to mobility independence. Therefore, physiotherapists should consider the outcome measures intended to examine lower limbs and the development of control between them.

In order to improve functional capacity and quality of life, Silva et al. [19] combined physiotherapy with PBM for patients with spinal cord injuries. The physiotherapeutic protocol consisted of stretching and muscle strengthening exercises. The group submitted to combined treatment (physiotherapy and PBM) presented more significant improvements in sensory and motor recovery than those submitted to physiotherapy alone. In another clinical trial, Silva et al. [20] found that PBM led to improvements in the motor responses of individuals with spinal cord injury, as demonstrated by differences in the EMG signals before and after treatment. These results motivated the present treatment protocol for children with MMC.

PBM has been used as an innovative modality to stimulate neural activity and the central nervous system in recent decades. This technique is based on the exposure of neural tissue to the low fluence of light at a wavelength in the red to an infrared range of the spectrum [33].

The effects of PBM have also been investigated in studies involving experimental spinal cord injury models. Wu et al. [34] demonstrated improvements in functional recovery and axonal regeneration following PBM at wavelengths of 808 and 700 nm administered daily for 14 days. In another experimental model, Paula et al. [35] demonstrated the positive effect of PBM at a wavelength of 780 nm (administered to five points in the region of the injury) on the recovery of the spinal cord, leading to faster functional recovery. Veronez et al. [36] found an improvement in tactile sensitivity following transcutaneous PBM at a wavelength of 808 nm when administered with 1000 J/cm^2^ to a single point in the region of the injury.

In the studies conducted by Silva et al. [18,19], individuals with spinal cord injuries received low-level laser irradiation at five points in the area of the injury. The PBM parameters were 808 nm, aperture diameter of 0.18 cm, 25 J per point, and exposure time of approximately 17 min. Sessions were held three times per week for four weeks, totaling 12 sessions. In the present study, an LED device was used with a wavelength of 808 nm, aperture diameter of 1.9 cm, 25 J per point, four points irradiated in the area of the lesion, and an approximate exposure time of three minutes.

A non-coherent light resource was used in the present study considering that LED devices show a higher bandwidth and flat matrices, which significantly increases the beam area, facilitating the treatment of large areas in less time. Another critical factor is that an LED device is less expensive than a laser device, making it more accessible [16,17]. A review study argues that optimal doses of PBM are related to different factors [37], and considering the penetration of lights, it is known that lasers penetrate deeper than LEDs [16]. Therefore, using a low-level laser in this condition and study context could result in more effective results considering the application locally.

## 5. Conclusions

In the present study, it was observed that LED did not present significant results regarding the evaluated items (hence what they were). On the other hand, we noticed important changes in the sensitivity; in this way, new studies using LED that focus on sensory evaluation should be performed. We believe that the combined use of these therapies can bring greater benefits to patients with myelomeningocele.

## Figures and Tables

**Figure 1 jcm-12-02920-f001:**
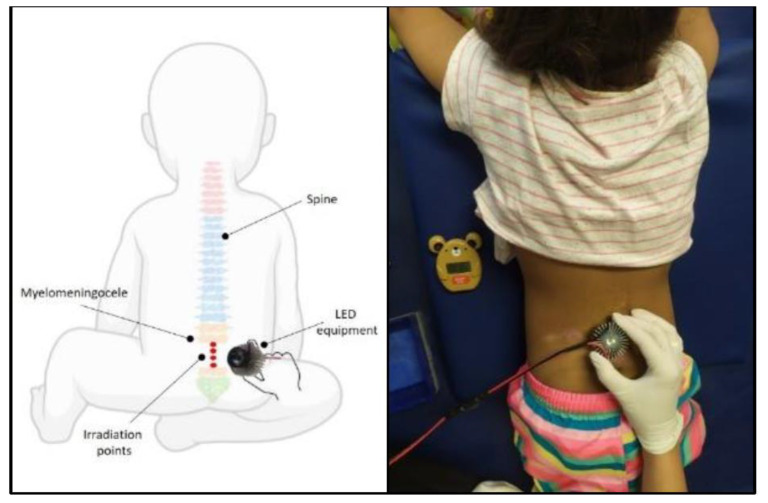
Positioning of participants for irradiation. Author’s personal collection.

**Figure 2 jcm-12-02920-f002:**
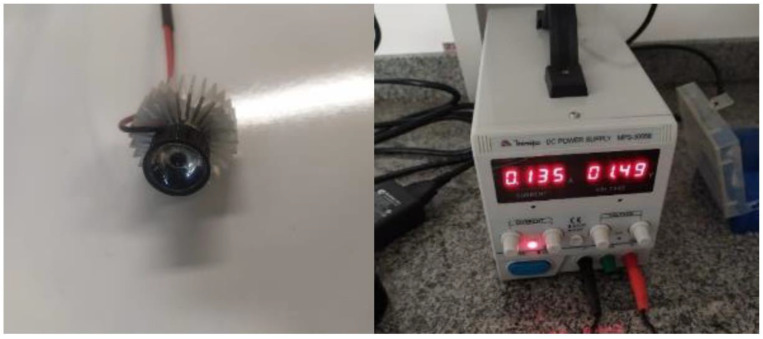
LED device. Author’s personal collection.

**Figure 3 jcm-12-02920-f003:**
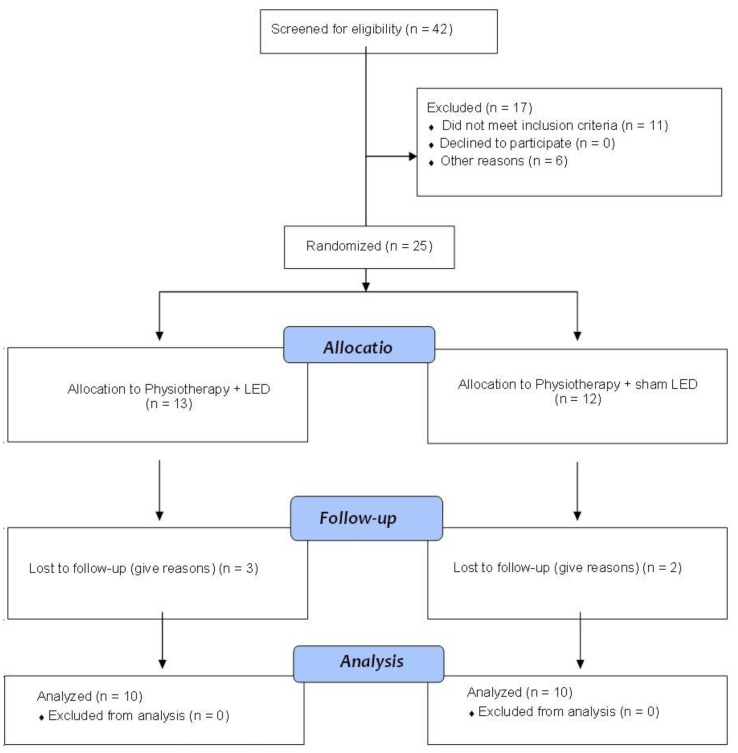
Flowchart.

**Figure 4 jcm-12-02920-f004:**
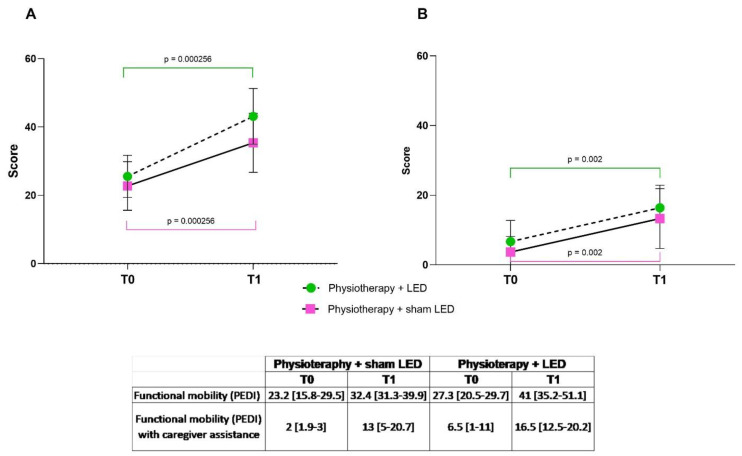
PEDI scores without (**A**) and with caregiver assistance (**B**)—T0: pre-intervention; T1: after 24 sessions. Data expressed as median and interquartile range. ** *p* < 0.01.

**Figure 5 jcm-12-02920-f005:**
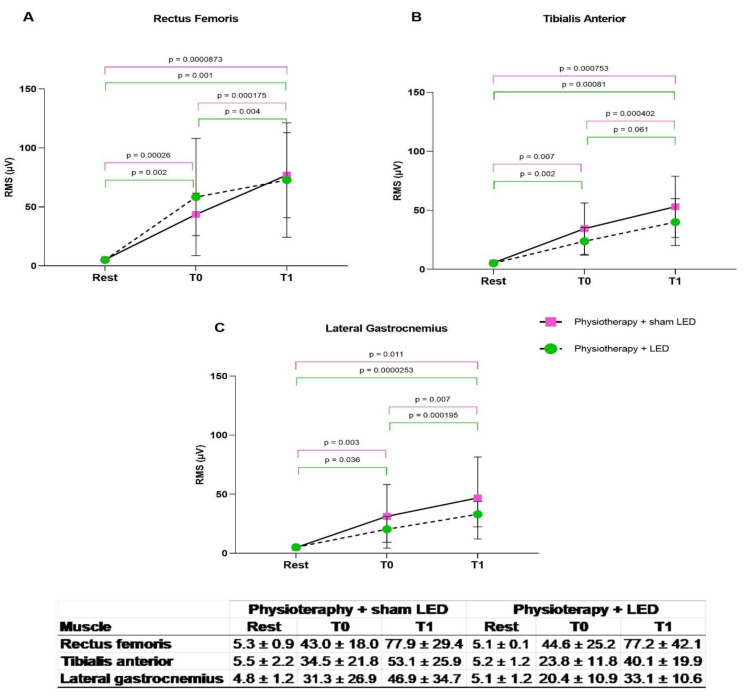
(**A**–**C**) RMS values of more compromised lower limb. Improvements in electrical muscle activity found in both treatment groups. T0 = pre-intervention; T1 = after 24 sessions; STS = sit-to-stand task.

**Figure 6 jcm-12-02920-f006:**
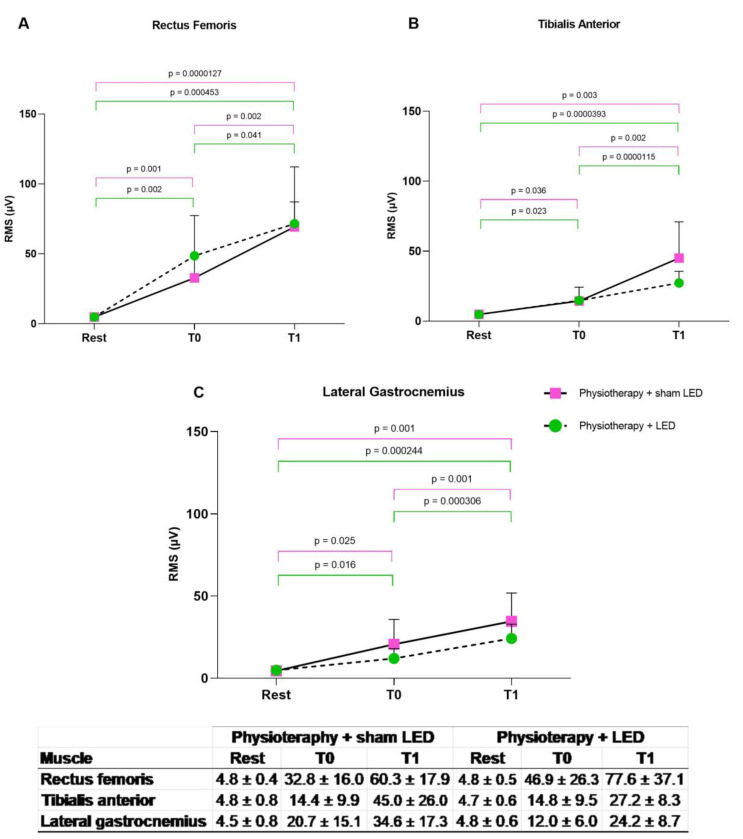
(**A**–**C**) RMS values of less compromised lower limb. Improvements in electrical muscle activity found in both treatment groups. T0 = pre-intervention; T1 = after 24 sessions; STS = sit-to-stand task.

**Table 1 jcm-12-02920-t001:** Parameters.

Parameter	
*Center wavelength [nm]*	850
*Spectral bandwidth (FWHM) [nm]*	20
*Operating mode*	*Continu* *ous wave*
*Average radiant power [mW]*	500
*Polarization*	*Random*
*Aperture diameter [cm]*	1.9
*Irradiance at aperture [mW/cm^2^]*	176
*Beam profile*	*Multimode*
*Beam spot size at target [cm^2^]*	2.84
*Irradiance at target [mW/cm^2^]*	176
*Exposure duration per point [s]*	50
*Radiant exposure [J/cm^2^]*	9
*Radiant energy per point [J]*	25
*Number of points irradiated*	4
*Radiant energy per session [J]*	100
*Area irradiated per session [cm^2^]*	11.34
*Application technique*	*Contact*
*Number of treatment sessions*	24
*Frequency of treatment sessions*	2× *per week*
*Total radiant energy [J]*	2400

**Table 2 jcm-12-02920-t002:** Characteristics of participants.

*Características*	*Physiotherapy + LED (n = 10)*	*Physiotherapy + Sham LED (n = 10)*
*Age*	3.8 ± 1.6	3.2 ± 1.6
*Sex (girls/boys)*	9/1	10/0
*Lesion level n (%)*		

*Upper lumbar*	0	0
*Lower lumbar*	10 (100%)	10 (100%)
*Sacral*	0	0
*Ambulator (yes/no)*	0/10	0/10
*Arnold Chiari malformation (yes/no)*	10/0	10/0
*Tethered spinal cord (yes/no)*	0/10	0/10
*Neurogenic bladder (yes/no)*	10/0	10/0
*Neurogenic intestine (yes/no)*	10/0	10/0

## Data Availability

All the data used to support the findings in this study are included in the article.

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
