# Peer review of "Effect of Photobiomodulation Combined with Physiotherapy on Functional Performance in Children with Myelomeningo-Cele-Randomized, Blind, Clinical Trial"

_jcm, 2023, doi:10.3390/jcm12082920_

Round 1

Reviewer 1 Report

The authors present their sudy on LED photobiomodulation in children with lumbar MMC and the typical neurological pattern of disease consisting in Chiari malformation, bladder and intestine malfunction, motor dysfunction of the lower limbs. After 12 weeks of study in two groups with physiotherapy and LED against physiotherapy and sham the results coud show improvements in both groups and no significant effects by LED.

Children with MMC and Chiari malformation are a highly heterogeneous group with different grades of involvement of the neural placode, glial scarring and  hypersensibility by axonal cross talk  to just mention some parts of the spectrum. LED therapy has been reported to induce nitric oxide mediated improvements of tissue perfusion and to modulate endothelial function. The present study is one of the first to test LED therapy in MMC. The number of recruted participants in each group were to small to get a significant result regarding the multifactorial disease pattern. Therefor the study has a preliminary character which does not reduce its importance and interest. 

Author Response

Manuscript ID: jcm-2231912

Title: EFFECTS OF PHOTOBIOMODULATION COMBINED WITH PHYSIOTHERAPY ON

FUNCTIONAL PERFORMANCE IN CHILDREN WITH MYELOMENINGOCELE – RANDOMIZED,

BLIND, CLINICAL TRIAL

Dear reviewers,

Thank you very much for the thorough revision of our manuscript. We are at your disposal for any further amendments and suggestions you may find necessary.

We respond to your comments carefully. We are available if further clarification is required.

The changes are also highlighted in yellow and bold in the revised manuscript.

Yours faithfully.

Reviewer 2 Report

Dear  authors:

The manuscript “Effect of Photobiomodulation Combined With Physiotherapy on Functional Performance in Children With Myelomeningo-cele-Randomized, Blind, Clinical Trial” It is a very interesting research and pertinent about the pertinent of this topic.

There is some questions about the manuscript, that can be usefull to improve the paper

Respect to the sections:

ABSTRACT

PURPOSE: Evaluate the effects of photobiomodulation (PBM) associated with physiother-apy on the functional performance of children with myelomeningocele.” The abstract structure of this journal indicates the that the PURPOSE may be can sustituite for “Backgraund”.

Introduction section

The bibliography may be cited in the text inside [ ] symbol.

There is no hypothesis included in the eintroduction section of  the research.

“Therefore, in this study we sought to evaluate the effects of physiotherapy combined with PBM on the functional performance of children with myelomeningocele as well as electrical activity of the rectus femoris, tibialis anterior and lateral gastrocnemius muscle during the sit-to-stand task.” Compared to? After which intervention? I think it not clear, the aim maybe can be rewrite.

Material and Methods

Wich ethical comitee approved the study, this is a very important question.

Participants (the age of the participants it is not homogeneus, and the sex) this aspect seems to be  possible limitation or sesgo. I can not see it very clear, maybe it is important to indicate it clearly? There was only a boy?

The randomization may be cited to two groups, experimental and control group to more understand to the readers I consider.

Experimental design

The participants were selected based on eligibility criteria and randomly allocated to either the Active LED group (n = 10) or Sham LED (n = 10). Both guardians were blinded to group allocation. Before and after the training protocol, the participants underwent measures of body structures, functional mobility and caregiver assistance.” Can be more explained what was considered about the individual intervention for each participant?

Figure 1 and figure 2 is it an own figure?

“The intervention was discontinued if the partici-pant missed two or three non-consecutive sessions.” If the participant missed to two or three consecutives sessions was discontinued? It is not clear 2 or 3 sessions? Maybe it need to be more agreed.

“Electrical muscle activity was first determined with the child sitting at rest. The child was then asked to stand, remain in the standing position for ten seconds and sit down again. The sit-to-stand task was performed three times with a five-minute interval be-tween trials. The children performed the movement at a pace usually adopted in their daily routine. Collection time was approximately 30 minutes, always with rest intervals between readings to avoid the effects of fatigue.” I think this protocol is a bit complicate to a 2 years child to understand.

Results

The flowchart data are not coherent respect to the excluded participants, Were 17 participants that were excluded, however only 11 not meet the inclusion criteria( Therefore 6 subjects are missing)

The references are not in the MDPI format.

Best regards

Author Response

ID do manuscrito: jcm-2231912

Tipo de manuscrito: Artigo

Título: EFEITOS DA FOTOBIOMODULAÇÃO COMBINADA COM FISIOTERAPIA SOBRE

DESEMPENHO FUNCIONAL EM CRIANÇAS COM MIELOMENINGOCELE – RANDOMIZADO,

CEGO, ENSAIO CLÍNICO

Caros revisores,

Muito obrigado pela revisão completa do nosso manuscrito. Estamos à sua disposição para quaisquer alterações e sugestões que julgar necessárias.

We respond to your comments carefully. We are available if further clarification is required.

The changes are also highlighted in yellow and bold in the revised manuscript.

Yours faithfully.

(Reviewer 2)

ABSTRACT

PURPOSE: Evaluate the effects of photobiomodulation (PBM) associated with physiother-apy on the functional performance of children with myelomeningocele.” The abstract structure of this journal indicates the that the PURPOSE may be can sustituite for “Backgraund”.

Author’s Response: Thank you for your comment. Please see the requested changes on line 27.

Introduction section

The bibliography may be cited in the text inside [ ] symbol.

Author’s Response: Thank you for your comment. We have added the symbol [].

There is no hypothesis included in the eintroduction section of  the research.

“Therefore, in this study we sought to evaluate the effects of physiotherapy combined with PBM on the functional performance of children with myelomeningocele as well as electrical activity of the rectus femoris, tibialis anterior and lateral gastrocnemius muscle during the sit-to-stand task.” Compared to? After which intervention? I think it not clear, the aim maybe can be rewrite.

Author’s Response: Thank you for your comment. We understand the importance of including the outlined information, and therefore we have reviewed the study objective and included a study hypothesis.

Material and Methods

Wich ethical comitee approved the study, this is a very important question.

Author’s Response: Thank you for your comment. Please refer to the method session, line 97, “Research Ethics Committee of University Nove de Julho (number: 4,308,134)”.

Participants (the age of the participants it is not homogeneus, and the sex) this aspect seems to be  possible limitation or sesgo. I can not see it very clear, maybe it is important to indicate it clearly? There was only a boy?

Author’s Response: Thank you for your comment. In agreement with reviewer’s suggestion, we have modified these accordingly. In the inclusion criteria session, we described that children between 2 and 12 of age would be included in our study. However, in table 1, we showed that participants from both groups had similar ages. In addition, only a male participant was in the study, as per table 1. 

The randomization may be cited to two groups, experimental and control group to more understand to the readers I consider.

Author’s Response:

Experimental design

The participants were selected based on eligibility criteria and randomly allocated to either the Active LED group (n = 10) or Sham LED (n = 10). Both guardians were blinded to group allocation. Before and after the training protocol, the participants underwent measures of body structures, functional mobility and caregiver assistance.” Can be more explained what was considered about the individual intervention for each participant?

Author’s Response: Thank you for your comment. The neurofunctional physiotherapy protocol was described in the intervention section for both intervention and control groups. Also, in the PMB section upon Highlighted in yellow, we defined treatment procedures for the placebo group and active PMB.

Figure 1 and figure 2 is it an own figure?

Author’s Response: Yes, that is correct. We also indicated that to ensure transparency.   

“The intervention was discontinued if the partici-pant missed two or three non-consecutive sessions.” If the participant missed to two or three consecutives sessions was discontinued? It is not clear 2 or 3 sessions? Maybe it need to be more agreed.

Author’s Response: Thank you for your comment, and we apologize for the missing information. We have included the correct details. “The intervention was discontinued if the participant missed three non-consecutive sessions.”

“Electrical muscle activity was first determined with the child sitting at rest. The child was then asked to stand, remain in the standing position for ten seconds and sit down again. The sit-to-stand task was performed three times with a five-minute interval be-tween trials. The children performed the movement at a pace usually adopted in their daily routine. Collection time was approximately 30 minutes, always with rest intervals between readings to avoid the effects of fatigue.” I think this protocol is a bit complicate to a 2 years child to understand.

Resposta do autor: Obrigado por seu comentário e entendemos suas preocupações sobre isso. No entanto, todos os participantes incluídos no estudo foram capazes de compreender as instruções da atividade. Na impossibilidade de fazê-lo, o participante seria excluído do estudo.

Resultados

Os dados do fluxograma não são coerentes em relação aos participantes excluídos, Foram 17 participantes que foram excluídos, porém apenas 11 não atendem aos critérios de inclusão (portanto faltam 6 sujeitos)

Resposta do autor: Obrigado pelo seu comentário. Pedimos desculpas pelo erro tipográfico. Atualizamos a figura relacionada.

As  referências  não estão no formato MDPI.

Resposta do Autor: Obrigado pela sua observação, fizemos esta alteração.

Reviewer 3 Report

EFFECTS OF PHOTOBIOMODULATION COMBINED WITH PHYSIOTHERAPY ON FUNCTIONAL PERFORMANCE IN CHILDREN WITH MYELOMENINGOCELE – RANDOMIZED, BLIND, CLINICAL TRIAL

The authors studied the effect of PBM+physiotherapy  in 13 children (finally 10 children) and   PBM sham+physiotherapy in 12 (also finally 10). PBM was carried out with a LED device at four points and twelve-week of PH (two weekly 45 – 60 min sessions). The authors used the Pediatric Evaluation of Disability Inventory (PEDI) and muscle activity. The authors reported greater independence in performing the tasks, requiring less assistance from their caregivers. More significant electrical activity was found.

The study many important major points to concern:

-The title is confusing: Title does not express the manuscript (MN). In MN, introduction focused on myelomeningocele and in the discussion on the Physiotherapy. Very little information about photobiomodulation.

Abstract:

-Without a clear hypothesis, it may be difficult to understand the purpose and significance of the study. Additionally, without objective data or statistical analysis, it is difficult to determine the validity and reliability of the results.

-Regarding the mention of "electrical activity," it is unclear what type of electrical activity is being referred to. Superficial electromyography (sEMG) is a common method used to measure muscle activity, but it is unclear if this is the type of electrical activity being measured in the study.

-Overall, it may be beneficial for the study authors to provide more detailed information about their methods and results to improve the clarity and validity of their study.

Introduction

-Without clear information about photobiomodulation, and without clear objective it is difficult to understand why the study was done.

Methods:

-How was Cognitive impairment evaluated?

-Electrical muscle activity: what did they measure? Latency? Amplitude? Area? How did they unify the experiment in children whose age was between 2-12 years?

Data analysis

-Should be given which data was parametric data, which was not.

Result:

-Tables with rough data will be very informative

-In spite of significant result, very small number of children was included. And there was big variability of age.

-The result of two-way ANOVA was not given completely.

Discussion

It focused on the physiotherapy, with very little information about photobiomodulation

Conclusion is not related with their result and discussion.

Figures:

The language should be edited which is in other language (Portugal?)

Author Response

ID do manuscrito: jcm-2231912

Tipo de manuscrito: Artigo

Título: EFEITOS DA FOTOBIOMODULAÇÃO COMBINADA COM FISIOTERAPIA SOBRE

DESEMPENHO FUNCIONAL EM CRIANÇAS COM MIELOMENINGOCELE – RANDOMIZADO,

CEGO, ENSAIO CLÍNICO

Caros revisores,

Thank you very much for the thorough revision of our manuscript. We are at your disposal for any further amendments and suggestions you may find necessary.

We respond to your comments carefully. We are available if further clarification is required.

The changes are also highlighted in yellow and bold in the revised manuscript.

Yours faithfully.

(Reviewer 3)

-The title is confusing: Title does not express the manuscript (MN). In MN, introduction focused on myelomeningocele and in the discussion on the Physiotherapy. Very little information about photobiomodulation.

(Reviewer 3)

Author’s Response: Thank you for your comment. We have made the requested changes accordingly. The introduction and discussion sections include further details on the physiotherapy approach in myelomeningocele and PMB.

Abstract:

-Without a clear hypothesis, it may be difficult to understand the purpose and significance of the study. Additionally, without objective data or statistical analysis, it is difficult to determine the validity and reliability of the results.

Author’s Response: We appreciate your feedback. we include the hypothesis in the introduction to the study. We do not put statistical values in the abstract due to the limit of words it must contain, it would be too long. We apologize.

-Regarding the mention of "electrical activity," it is unclear what type of electrical activity is being referred to. Superficial electromyography (sEMG) is a common method used to measure muscle activity, but it is unclear if this is the type of electrical activity being measured in the study.

Author’s Response: Thank you for your comment. We have added the following information to the study abstract: “The RMS data were extracted and analyzed.”

-Overall, it may be beneficial for the study authors to provide more detailed information about their methods and results to improve the clarity and validity of their study.

 Author’s Response: Thank you for your comment. We have reviewed both methods and results sections accordingly to ensure transparency in our findings.

Introduction

-Without clear information about photobiomodulation, and without clear objective it is difficult to understand why the study was done.

Author’s Response: Thank you for your comment. We have added further information about physiotherapy and PBM in the introduction section.

Methods:

-How was Cognitive impairment evaluated?

Author’s Response: Thank you for your comment. We did not evaluate cognitive impairment in our cohort. However, one of the study's inclusion criteria was understanding instructions for sitting and standing tasks during EMG measures. This is described in the 'Inclusion and Exclusion Criteria' section.

-Electrical muscle activity: what did they measure? Latency? Amplitude? Area? How did they unify the experiment in children whose age was between 2-12 years?

Author’s Response: Thank you for your comment. To be included in the study, children should be between 2 and 12 years of age. However, participants did not have significant age differences, as described in Table 1. Regarding EMG, we evaluated the RMS. A very common way of representing the intensity or amplitude of the electromyographic signal is produced by calculating the root mean square (RMS), which is an average calculated by the computerized electromyograph program that represents the voltage along the study cycle. It is the RMS of the input values. If N is, the number of samples and xi is the value of each sample. We include this information in the lines:

Data analysis

-Should be given which data was parametric data, which was not.

Author’s Response: Thank you for your comment. Information on parametric and non-parametric tests can be found in the methods section, particularly the Statistical Analysis item.

Result:

-Tables with rough data will be very informative

Author’s Response: Thank you for your comment. The results figures include p-value, mean, standard deviation and interquartile range information.

-In spite of significant result, very small number of children was included. And there was big variability of age.

Author’s Response: Thanks for that comment. We agreed that the sample size was small, yet sufficient for statistical analysis. There was no significant variation concerning the participants' age. Please note that only the inclusion criteria described that children from 2 to 12 years old could be included in the study. Information on their average age can be found in table 1, demonstrating no major differences in age.

-The result of two-way ANOVA was not given completely.

Não entendi essa questão. My apologies but could not understand the question.

Discussion

It focused on the physiotherapy, with very little information about photobiomodulation

Author’s Response: In agreement with the reviewer's suggestion, we have added a paragraph approching the PBM in the discussion section.

A non-coherent light resource has been chosen in the present study considering that LED devices show a higher bandwidth and flat matrices, which significantly increase the beam area, facilitating the treatment of large areas in less time. Another critical factor is that an LED device is less expensive than a laser device, making it more accessible [25]. A review study argues that optimal doses of PBM are related to different factors [36] and considering the penetration of lights, it is known that lasers penetrate deeper than LEDs [16]. Therefore, the use of a low-level laser in this condition and study context could result in more effective results considering the application locally”.

Conclusion is not related with their result and discussion.

Resposta do autor: Fizemos as alterações solicitadas na discussão e na introdução.

Figuras:

O idioma deve ser editado que é em outro idioma (Portugal?)

Resposta do autor: Obrigado por essa observação. Pedimos desculpas por este erro e alteramos o idioma desta figura.

Round 2

Reviewer 2 Report

Dear authors, thank you for considerate our questions

The manuscript has been improved

Reviewer 3 Report

no comment.